# On the Sensitivity of Standardized-Precipitation-Evapotranspiration and Aridity Indexes Using Alternative Potential Evapotranspiration Models

**Aristoteles Tegos [1,2,*], Stefanos Stefanidis [3], John Cody [2] and Demetris Koutsoyiannis [1]**

1   Laboratory of Hydrology and Water Resources Development, School of Civil Engineering, National Technical University of Athens, Heroon Polytechneiou 9, 15780 Zographou, Greece
2   Ryan Hanley Ltd. Ireland, 170/173 Ivy Exchange, Granby Pl, Parnell Square W, D01 N938 Dublin, Ireland
3   Laboratory of Mountainous Water Management and Control, School of Forestry and Natural Environment, Aristotle University of Thessaloniki, 54124 Thessaloniki, Greece
*   Correspondence: tegosaris@yahoo.gr

**Abstract:** This paper examines the impacts of three different potential evapotranspiration (PET) models on drought severity and frequencies indicated by the standardized precipitation index (SPEI). The standardized precipitation-evapotranspiration index is a recent approach to operational monitoring and analysis of drought severity. The standardized precipitation-evapotranspiration index combines precipitation and temperature data, quantifying the severity of a drought as the difference in a timestep as the difference between precipitation and PET. The standardized precipitation-evapotranspiration index thus represents the hydrological processes that drive drought events more realistically than the standardized precipitation index at the expense of additional computational complexity and increased data demands. The additional computational complexity is principally due to the need to estimate PET within each time step. The standardized precipitation-evapotranspiration index was originally defined using the Thornthwaite PET model. However, numerous researchers have demonstrated the standardized precipitation-evapotranspiration index is sensitive to the PET model adopted. PET models requiring sparse meteorological inputs, such as the Thornthwaite model, have particular utility for drought monitoring in data scarce environments. The aridity index (AI) investigates the spatiotemporal changes in the hydroclimatic system. It is defined as the ratio between potential evapotranspiration and precipitation. It is used to characterize wet (humid) and dry (arid) regions. In this study, a sensitivity analysis for the standardized precipitation-evapotranspiration and aridity indexes was carried out using three different PET models; namely, the Penman–Monteith model, a temperature-based parametric model and the Thornthwaite model. The analysis was undertaken in six gauge stations in California region where long-term drought events have occurred. Having used the Penman–Monteith model as the PET model for estimating the standardized precipitation-evapotranspiration index, our findings highlight the presence of uncertainty in defining the severity of drought, especially for large timescales (12 months to 48 months), and that the PET parametric model is a preferable model to the Thornthwaite model for both the standardized precipitation-evapotranspiration index and the aridity indexes. The latter outcome is worth further consideration for when climatic studies are under development in data scarce areas where full required meteorological variables for Penman–Monteith assessment are not available.

**Keywords:** drought; standardized precipitation-evapotranspiration index; aridity index; parametric PET model; California

## 1. Introduction

Drought is a severe natural hazard characterized by lower-than-normal precipitation that when extended over seasonal or longer timescales results in water resources that are in-

sufficient to meet the needs of human activities or environmental demands [1]. Aridity is a permanent characteristic of a region's climate regime; droughts are temporary phenomena. While a normal part of the climate regime, the spatial extent and severity of a drought event will vary on seasonal and annual timescales, especially in arid and semi-arid regions. Unlike other natural hazards, there is not a single universally accepted definition of a drought, and numerous drought indexes have been proposed over recent decades to provide a standardized definition of the phenomenon. The phenomenon has caused negative impacts on numerous natural, human and social activities and, over the past decades, numerous modelling approaches have been developed to quantify the severity of the phenomenon. Modelling choices comprise approaches using meteorological, hydrological, agricultural, comprehensive, remote sensing-based and combined drought indexes [2,3]. A detailed historical background on the drought indexes is presented by Heim (2002) [4], where the reader can find the historical evolution of drought indexes from rainfall-based approaches to more complex models that combine precipitation with evapotranspiration that seek to describe the phenomenon in terms of the relevant processes of the hydrological cycle. The standardized precipitation index (SPI) is widely used to characterize meteorological drought over a range of timescales. The standardized precipitation index quantifies precipitation as a standardized departure from a selected probability model. A limitation of the standardized precipitation index is that sensitivity to soil moisture, groundwater and reservoir stores is a function of the timescale selected; hence, the severity of droughts indicated by the standardized precipitation index is dependent on the timestep rather than solely on hydrological and climate processes.

The standardized precipitation-evapotranspiration index is a recent drought index [5]. It was first introduced as a multi-timescale (from monthly to 36 months) index incorporating rainfall and potential evapotranspiration models, with the latter variable modelled using a Thornthwaite temperature-based model. The standardized precipitation-evapotranspiration index appears to be a robust climate-meteorological drought index when considering its application globally over the last decade [6–10]. By integrating the potential evapotranspiration with the precipitation, the standardized precipitation-evapotranspiration index enables more advanced identification of drought types and drought impacts on diverse climatic systems than using only precipitation as a drought explanatory variable.

The aridity index describes the long-term functioning of the atmosphere; more specifically, it is the process of receiving and releasing water from the underlying surface hydrological system [11]. The index is also known as the Budyko index [12]. The standardized precipitation-evapotranspiration index [5] was introduced as a simplified drought monitoring tool for use by national agencies in detecting droughts.

Numerous authors have reported, in global evaluations of applications, that the standardized precipitation-evapotranspiration index was the adopted model that was most sensitive to estimate the potential evapotranspiration (PET) variable within the SPEI assessment [13–15]. The Thornthwaite PET model, despite its structural simplicity, fails to provide accurate results when used as an input to the standardized precipitation-evapotranspiration index. Ortiz-Gómez et al. (2022) [16] stated that drought events detected with the standardized precipitation-evapotranspiration index are more intense when the Thornthwaite model is used to calculate evapotranspiration instead of the Hargreaves–Samani model. Ogunrinde et al. (2020) [17] concluded that the standardized precipitation-evapotranspiration index estimated using the Hargreaves PET model (SPEI-H) and the standardized precipitation-evapotranspiration index estimated using the Penman–Monteith (SPEI-P) show higher correlation for all timesteps than the SPEI-Thornthwaite model. Shi et al. (2020) found that differences in projected increases of drought frequencies were found among the different PET models estimated by machine learning-based methods [18] when the standardized precipitation-evapotranspiration index was assessed.

Because of the sensitivity of the standardized precipitation-evapotranspiration index to PET, numerous researchers have investigated simplified assessments of PET over the past

eight decades, mainly focusing on developing parsimonious PET models that can quantify evapotranspiration. The Penman–Monteith model, which is the most well-established PET model, is complex to apply due to high input data demands [19]. The model requires temperature, radiation, wind velocity and humidity for its development [20].

The aim of this study is to provide new insights into the sensitivity of the selected potential evapotranspiration inputs for assessing the drought severity in the context of the standardized precipitation-evapotranspiration and aridity indexes; associated deficiencies can appear when undefined PET inputs are used within the context of drought index assessment. The PET estimation is a complex technical task within the hydrology domain, which strongly depends on the availability of numerous meteorological variables, leading to the use of simplified PET models, which usually do not achieve physical-interpretation outcomes. From this perspective, three PET models with different composite level were applied and compared; namely, Penman–Monteith, Thornthwaite and parametric, while using the Penman–Monteith model as the base model in six meteorological stations for a long-term period of 30 years (1982–2013). California is an area that has suffered from a long history of severe droughts [21] and is ideal for developing and testing drought related research.

## 2. Materials and Methods

### 2.1. Study Area

Full raw measurements from six meteorological stations across California (Figure 1) for the period 1983–2013 (Table 1) have been gathered. For the selected stations, the full monthly timeseries containing length of temperature, relative humidity, radiation and wind velocity has been available for a 30-year period and therefore has been mobilized for our study instead of using other CIMIS gauges stations with insufficient and incomplete meteorological records. The California Irrigation Management Information System (CIMIS) is a program unit in the Water Use and Efficiency Branch, Division of Regional Assistance, California Department of Water Resources (DWR), and it is one of the foremost global meteorological networks associated with potential evapotranspiration research. Historically, the network has been developed in cooperation with UC Davis. The local environmental and soil conditions in the meteorological stations allow accurate estimation of PET.

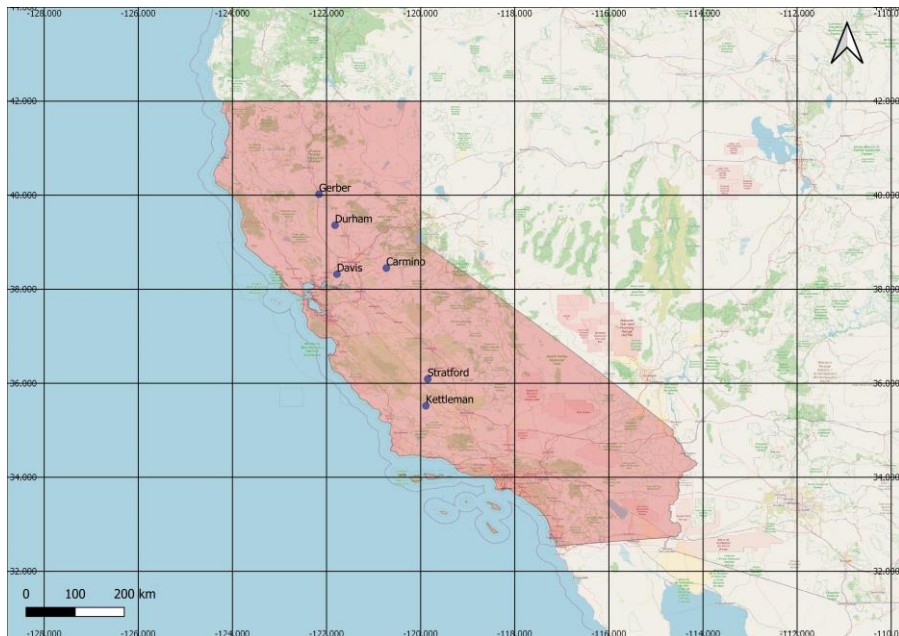

**Figure 1.** Study area and meteorological stations.

**Table 1.** Information on the selected stations.

| Sequence Number | Name | Meteorological Variables | Temporal Resolution | Time Period |
|---|---|---|---|---|
| 1 | Davis | Temperature, relative humidity, radiation, wind velocity | monthly | 1982–2013 |
| 2 | Gerber | Temperature, relative humidity, radiation, wind velocity | monthly | 1982–2013 |
| 3 | Durham | Temperature, relative humidity, radiation, wind velocity | monthly | 1982–2013 |
| 4 | Carmino | Temperature, relative humidity, radiation, wind velocity | monthly | 1982–2013 |
| 5 | Stratford | Temperature, relative humidity, radiation, wind velocity | monthly | 1982–2013 |
| 6 | Kettleman | Temperature, relative humidity, radiation, wind velocity | monthly | 1982–2013 |

*2.2. Modelling Procedures*

Raw monthly meteorological timeseries were acquired for six gauge stations. Four meteorological variables were collected; namely, mean temperature, radiation wind velocity and relative humidity. PET was estimated monthly by using three different models; namely, Penman–Monteith, parametric and Thornthwaite, and they are presented below with representative equations for each PET model:

The well-known Penman–Monteith [22] equation for estimating PET is expressed as:

$$\text{PET} = \frac{\Delta}{\Delta + \gamma'}\frac{R_n}{\lambda} + \frac{\Delta}{\Delta + \gamma'}F(u)D, \quad \gamma' = \gamma\left(1 + \frac{r_s}{r_a}\right)$$

where PET is potential evapotranspiration (mm/d), $R_n$ is net radiation at the surface (Wm$^{-2}$), $\Delta$ is the slope of the saturation vapor pressure curve (PaK$^{-1}$), $\gamma$ is the psychometric coefficient (=67 PaK$^{-1}$), while $r_s$ (s/m) and $r_a$ (s/m) are the surface and aerodynamic resistance factors, respectively.

The Thornthwaite model [23] is a well-established simplified temperature-base model. The model's form is:

$$\text{PET} = 1.6L_d\left(\frac{10T_a}{I}\right)^a$$

where PET is potential evapotranspiration (mm/d), $L_d$ is the daytime length, $T_a$ is the mean monthly air temperature (°C), $I$ the annual heat index and $\alpha$ is an empirically determined parameter which is a function of $I$.

The PET parametric model is a modern temperature-base model based on a simplification of the Penman–Monteith model. The model was first introduced in CIMIS gauge stations [24]. Recently, global parametric PET approaches have been presented based on global gauge stations [25] as well as in conjunction with advanced remote sensing temperature datasets [26]. The parametric model's expression is:

$$\text{PET} = \frac{aR_a - b}{1 - cT_a}$$

where PET (mm) is potential evapotranspiration, $R_a$ (KJm$^{-2}$) is extraterrestrial shortwave radiation and $T\alpha$ (°C) is the mean air temperature. The model contains three parameters; namely, $\alpha$ (kg kJ$^{-1}$), $b$ (kg m$^{-2}$) and $c$ (°C$^{-1}$), which are estimated through local calibration of Penman–Monteith data in a process described by Tegos et al. (2015) [24].

In the standardized precipitation-evapotranspiration index, described by Vicente-Serrano et al. [5], the water deficit of rainfall and the potential evapotranspiration are considered at different timescales from 1 month to 48 months. The standardized precipitation-evapotranspiration index uses a three-parameter log-logistic distribution to capture the

deficit between rainfall and potential evapotranspiration. As part of this study, the following calculation procedure was applied:

1.  The water balance was estimated using the following equation for three different PET models:

$$D_n^k = \sum_{i=0}^{k-1} P_{n-1} - \text{PET}_{n-1}$$

where $P$ is the rainfall (mm), PET is the potential evapotranspiration (mm), $k$ is the timescale (months) of the aggregation and $n$ is the calculation month.

2.  Drought classification (Table 2) is estimated by fitting in the empirical distribution $D$ as proposed by Koutsoyiannis and Montanari (2022) [27]. They employed three log-logistic distributions.

**Table 2.** The severity and values of SPEI.

| Drought Category | SPEI Value |
|---|---|
| No drought | >−0.5 |
| Mild drought | −0.5~−1 |
| Moderate drought | −1~−1.5 |
| Severe drought | −1.5~−2 |
| Extreme drought | <−2 |

The aridity index (AI) is a numerical indicator of the degree of dryness of the climate at a given location. In this study, a well-known formula was used to estimate the aridity index expressed as follows [28]:

$$\text{AI} = \frac{P}{\text{PET}}$$

where $P$ is the average annual precipitation (mm) and PET is the potential evapotranspiration (mm).

Herein, the PET required for the aridity index estimation is calculated by applying three methods. These methods are (a) Penman–Monteith, (b) Thornthwaite and (c) a parsimonious parametric approach based on a simplification of the Penman–Monteith formula. Additionally, the aridity index threshold values for climatic aridity zone classification were applied according to the UNESCO [29] and UNEP [30] classification schemes for the Penman and Thornthwaite-based methods, respectively (Table 3).

**Table 3.** Aridity index (AI) threshold values for climatic aridity zone classification according to UNESCO and UNEP classification schemes.

| | UNESCO (Penman) | UNEP (Thornthwaite) |
|---|---|---|
| Aridity Climate Zone | AI values | |
| Hyper-arid | <0.03 | <0.05 |
| Arid | 0.03–0.2 | 0.05–0.2 |
| Semi-arid | 0.2–0.5 | 0.2–0.5 |
| Dry sub-humid | 0.5–0.75 | 0.5–0.65 |
| Humid | >0.75 | >0.65 |

## 3. Results

### 3.1. SPEI in Davis Gauge Station

Figure 2 presents SPEI for timescales from 1 month to 24 months and for three different PET models using Davis gauge station It can be concluded that:

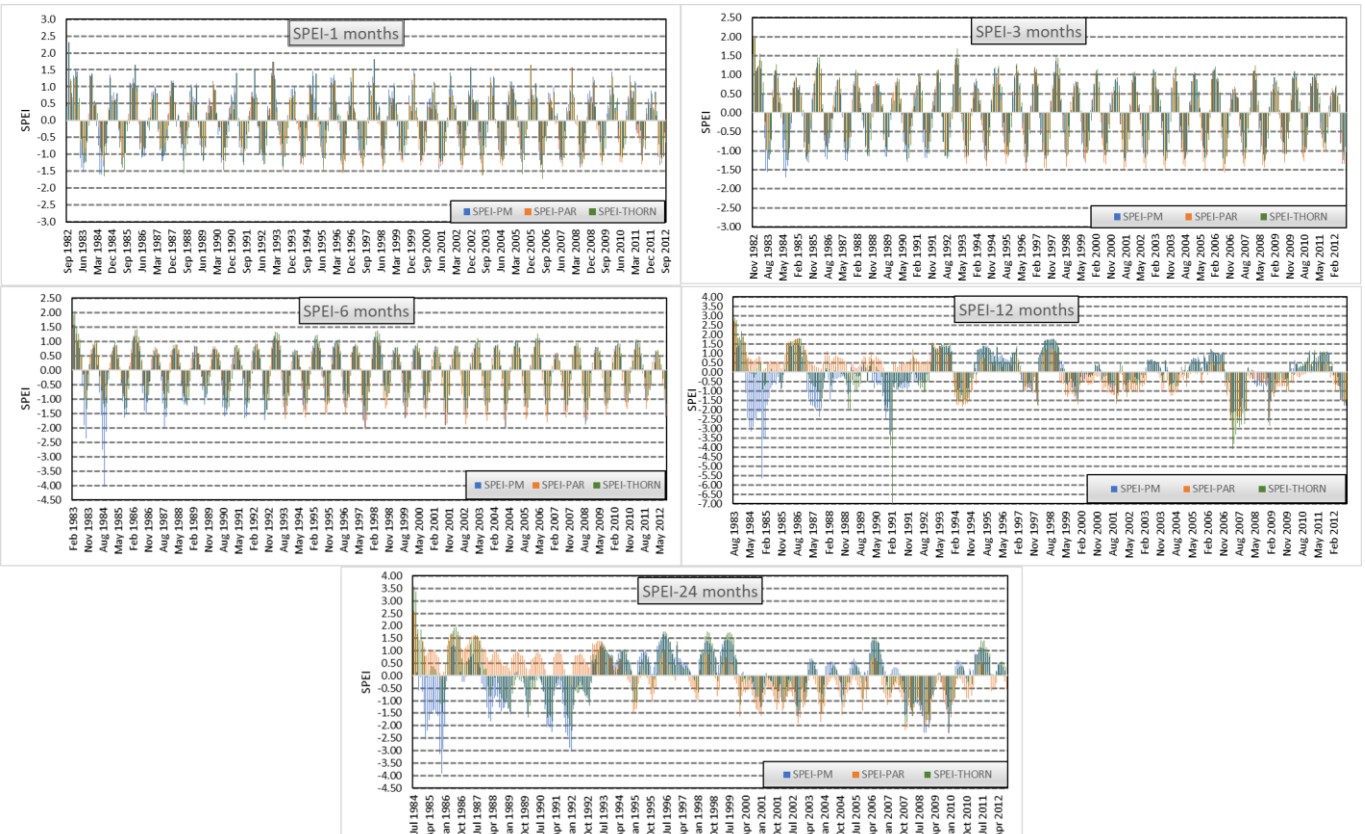

**Figure 2.** SPEI Davis meteorological station.

- Severe to extreme droughts are observed for the periods 1984–1985 and 1991–1992 for SPEI for 6-month and 12-month timescales. Lower SPEI timescales (1 month to 3 months) show annual occurrence of mild to moderate drought conditions. Limited SPEI values for 1984 are slightly lower than −1.5, which is a cut-off referring to severe drought conditions.
- For scales up to 6 months, the droughts class severity is underestimated by both parametric and Thornthwaite models. The latter presents the highest deficiencies against drought classes when compared to the parametric model.
- A major drought event (1984–1986) seems to be underestimated substantially by both PET models for the 12-month and 24-month timescales. A moderate drought event for the period 1987–1988 seems to be underestimated by the parametric model and less so by the Thornthwaite model.
- Overall, the consideration of alternative PET models proves the sensitivity of the drought classification when SPEI is analyzed.

Figure 3 shows common SPEI plots for all three PET models in reverse recorded order sequence. Following the above observations, it can be noted that:

- All three PET models provide similar drought SPEI classification up to 3-month timescales.
- The Thornthwaite model underestimates drought severe class in some 6-month events whereas the parametric model provides a more accurate classification of those events.
- The Thornthwaite model overestimates drought classes for 12-month events while the parametric model does the opposite.
- Both PET models (parametric, Thornthwaite) overestimate 24-month drought classes.

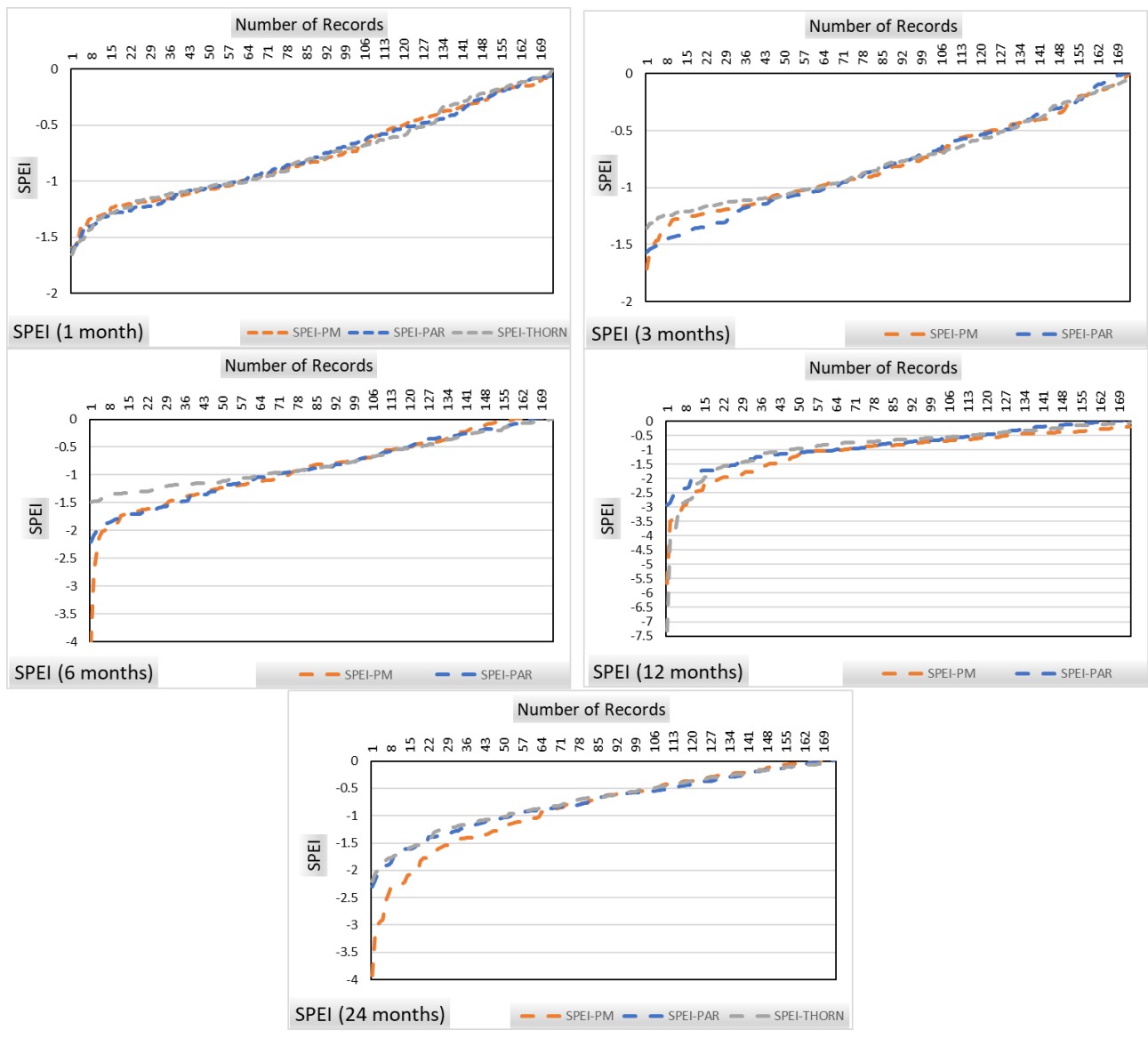

**Figure 3.** SPEI drought empirical plots (Davis station.

### 3.2. Error Analysis of the Total Sample

To consolidate our understanding on the influence of different PET configurations for quantifying standardized precipitation-evapotranspiration index classifications, standardized precipitation-evapotranspiration index values of different PET models were plotted for the total sample of five gauge stations; namely Gerber, Durham, Carmino, Stratford and Kettleman.

Figure 4 shows the common plots using the SPEI-Penman–Monteith (SPEI-PM) index as a base for evaluating the performance of both SPEI-parametric (SPEI-PAR) and SPEI-Thornthwaite (SPEI-THORN) indexes. Following this, it can be concluded that:

- SPEI-parametric and SPEI-Thornthwaite for up to 6 months provide similar drought classifications to the SPEI-PM index. The SPEI-parametric index shows better fit when compared to the SPEI-PM indexes. The latter proves that the PET parametric model has better performance than the PET Thornthwaite model.
- Drought severity is underestimated at 12 months by both SPEI-Thornthwaite and FSPEI-parametric models for limited drought events.
- A high classification variance is observed for SPEI at 24 months with both underestimating drought severity with SPEI-parametric and overestimating drought severity with SPEI-Thornthwaite.

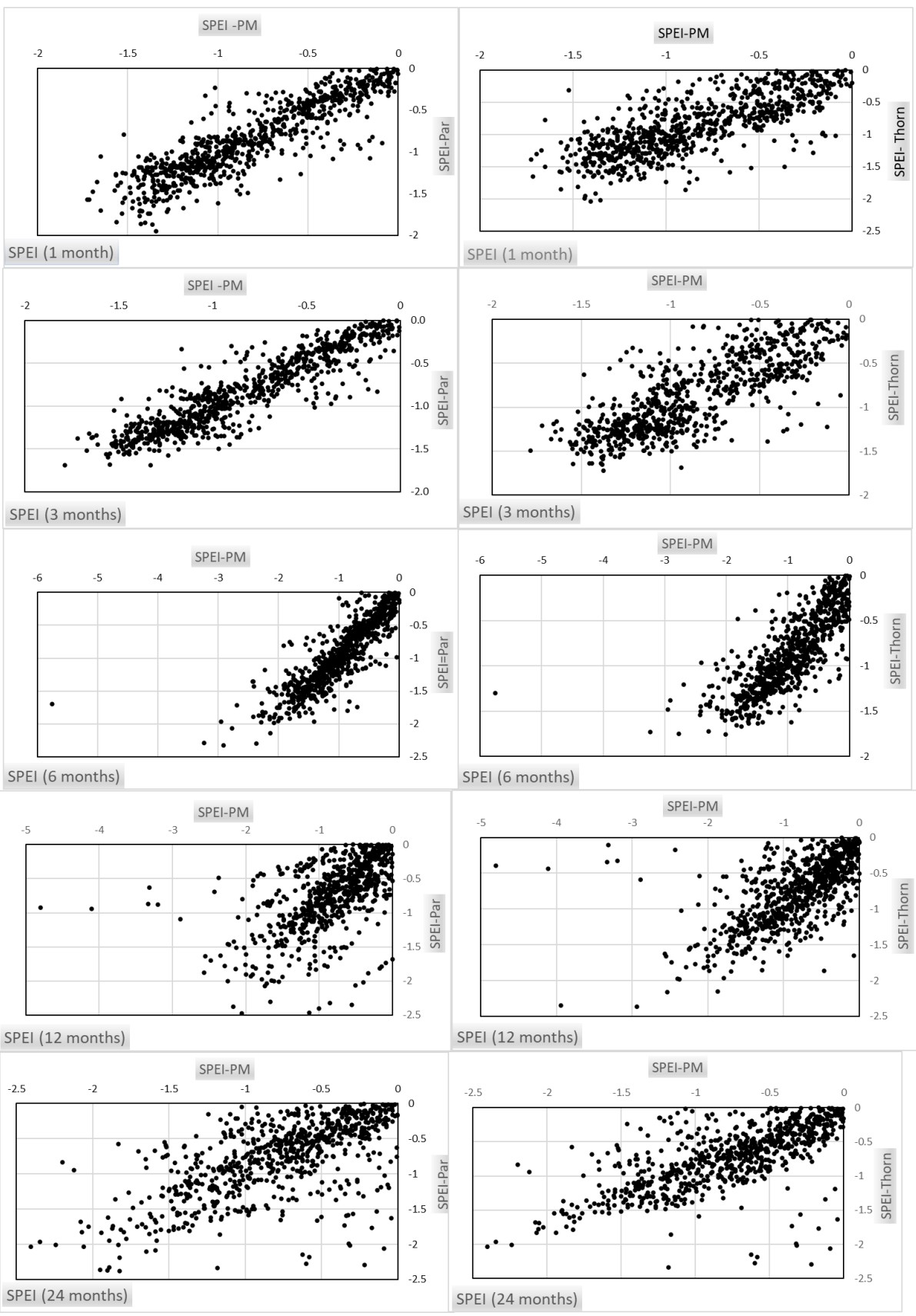

**Figure 4.** SPEI drought empirical plots (samples from five gauge stations).

### 3.3. Error Analysis of the Aridity Index

Based on the data from six CIMIS stations, monthly PET was calculated using the well-known Thornthwaite model and the above-mentioned modified parametric method for a normal 30-year climatic period (1983–2012). In addition, the calculated monthly Penman–Monteith PET timeseries served as the reference dataset for comparisons between the two different methods. Figure 5 illustrates the mean annual PET values derived from the reference and the other methods at each station.

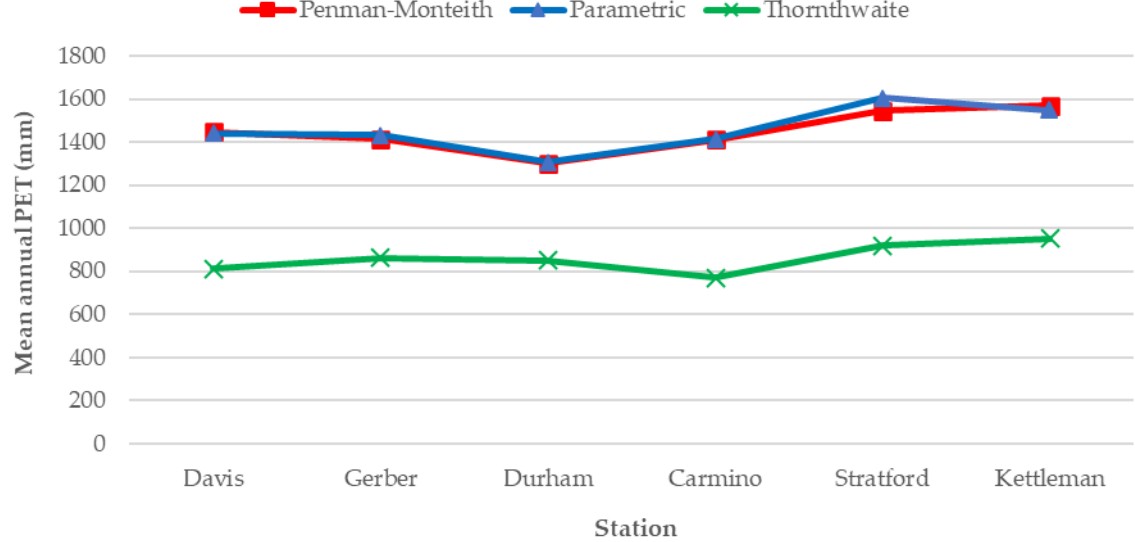

**Figure 5.** Mean annual Penman–Monteith PET against the parametric and Thornthwaite methods at each station.

It is clear that the parametric model is superior, while a significant underestimation has been observed by the Thornthwaite's method. More details on the limitations of alternative PET models are presented in the discussion section. This is also confirmed by the statistical metrics used herein to assess the performance of each method (Table 4).

**Table 4.** Values of the statistical indexes used to evaluate the performance of the (a) parametric and (b) Thornthwaite methods in the assessment of annual PET.

| Station | RMSE (mm) | | MBE (mm) | | Correlation (r) | |
|---|---|---|---|---|---|---|
| | Parametric | Thornthwaite | Parametric | Thornthwaite | Parametric | Thornthwaite |
| Davis | 67.6 | 642.6 | −6.2 | −637.7 | 0.57 | 0.27 |
| Gerber | 103.8 | 560.6 | 16.0 | −551.9 | 0.31 | 0.1 |
| Durham | 64.2 | 458.7 | 5.5 | −450.5 | 0.58 | 0.31 |
| Carmino | 65.1 | 643.3 | 3.2 | −640.0 | 0.64 | 0.63 |
| Stratford | 94.7 | 638.6 | 29.1 | −630.9 | 0.61 | 0.1 |
| Kettleman | 75.2 | 618.7 | −23.8 | −615.9 | 0.56 | 0.73 |

The findings show that the average RMSE is 78.4 mm and 593.8 mm for the parametric and Thornthwaite methods, respectively. Regarding the parametric method, the RMSE values range from 67.6 to 103.8 mm, while the RMSE values for the Thornthwaite method range from 458.7 to 643.3 mm. On the other hand, the average MBE for the parametric method range from −23.8 to 3.2, while considerably larger values were estimated for the Thornthwaite method, ranging from −450.5 to −640 mm. It is also notable that both methods present low correlation, as the average correlation coefficient is 0.55 for the parametric method and 0.36 for the Thornthwaite method.

Uncertainties introduced by the use of different PET methods highly affect the estimation of the aridity index. This may lead to an incorrect climatic classification at a given location (Table 5).

**Table 5.** Uncertainties in climate zone characterization due to the PET method.

| | Penman–Monteith | | Parametric | | Thornthwaite | |
|---|---|---|---|---|---|---|
| **Station** | **AI Value** | **Climatic Zone** | **AI Value** | **Climatic Zone** | **AI Value** | **Climatic Zone** |
| Davis | 0.320 | Semi-arid | 0.321 | Semi-arid | 0.571 | Sub-humid |
| Gerber | 0.465 | Semi-arid | 0.460 | Semi-arid | 0.762 | Sub-humid |
| Durham | 0.454 | Semi-arid | 0.452 | Semi-arid | 0.695 | Sub-humid |
| Carmino | 0.662 | Sub-humid | 0.660 | Sub-humid | 1.212 | Humid |
| Stratford | 0.133 | Arid | 0.128 | Arid | 0.224 | Semi-arid |
| Kettleman | 0.155 | Semi-arid | 0.158 | Semi-arid | 0.256 | Sub-humid |

According to the estimated aridity index using the reference PET method (Penman–Monteith), the climate was classified as semi-arid in four of the six stations, arid in one and sub-humid in one. A successful climatic zone classification was also achieved using the parametric PET method as an input parameter in the aridity index formula. It is also notable that the numerical values of aridity index between the two approaches are almost equal. On the contrary, the use of the Thornthwaite's PET method leads to significant errors in aridity index values. These differences factor into the climate dryness underestimation.

## 4. Discussion

Herein, we discuss how our analysis can contribute to existing knowledge of drought analysis by summarizing some key points of future research and issues for further consideration:

1. Potential evapotranspiration is the most complex meteorological process and significant numbers of simultaneous meteorological variables are required for its indirect estimation. The importance of simplified PET models is noteworthy. In this vein, the recent temperature-base parametric model can support drought studies when full meteorological gauges for estimating using the Penman–Monteith model are not available. As highlighted above parametric, the PET model outperforms the Thornthwaite PET model when the standardized precipitation-evapotranspiration index is assessed. The Thornthwaite PET model fails to provide accurate PET estimates, especially in arid and semi-arid areas and seems to be suitable for use only in warm climates where the temperature is the main PET driver.

2. The parametric PET model is recommended for use throughout the majority of the Earth in both arid and humid environments. Further research for improving the model's performance is proposed in tropical and sub-tropical environments, as is detailed by Tegos et al. (2017) [25] and dos Santos et al. (2021) [31].

3. From previous studies, the parametric PET model, even though it is robust, tends to underestimate monthly summer PET peaks, and monthly summer PET peaks may impact the drought severity during water stressed seasons. Thus, the development of a PET stochastic model can provide further insights in drought studies [32,33] if a stochastic component is considered and embedded within a parsimony framework as set out in previous studies [34].

4. Drought assessment and forecast remains a challenging task. In line with the advanced development of global remotely sensed models, new long term satellite datasets can support further drought assessment [35,36]

5. New advanced computational tools associated with different drought indexes are necessary to support geoscientists to capture a holistic view of the phenomenon [37].

6. Simple index approaches associated with top-down models have received criticism when the drought classification is considered under the short-term with a lack of gauge records. The recent development of multidimensional machine learning–

based algorithms may provide opportunities for developing drought forecasting models [38–40]. Comparative analysis among the different drought indexes is also recommended in order to improve our knowledge on the drought natural hazard as a natural phenomenon [41], since the definition of the drought among different scientific disciplines remains challenging [42].

## 5. Conclusions

This study introduced a sensitivity analysis of the potential evapotranspiration inputs based on the standardized precipitation-evapotranspiration index (SPEI) and aridity index frameworks. Three PET models of different complexity have been used; namely the Penman–Monteith model, the temperature-based parametric model and the empirical Thornthwaite model. The Penman–Monteith model was used as the base model for assessing the drought severity based on the standardized precipitation-evapotranspiration index definition. It can be summarized that for up to 6 months, the standardized precipitation-evapotranspiration index–parametric model and the standardized precipitation-evapotranspiration index–Thornthwaite model provide similar drought classifications, and that with the standardized precipitation-evapotranspiration index–PM model, drought severity is underestimated by both the standardized precipitation-evapotranspiration index–Thornthwaite model and the standardized precipitation-evapotranspiration index–parametric model for up to 12 and 24 months. In aridity index analysis, high uncertainties were introduced using different PET methods, which may lead to incorrect climatic classifications at different locations. Our research was carried out in the CIMIS network California, a semi-arid area with a long record of drought events; its gauge network is based on the local conditions, which allows estimation of PET with high reliability. It should be noted that both the SPEI and the aridity index are sensitive to drought classification when different PET models are used. The parametric model, which is a parsimonious model approximating the Penman–Monteith model, proved to be a valuable alternative when a full meteorological dataset for the Penman–Monteith model is not available, as it outperforms the simplified Thornthwaite model when standardized precipitation-evapotranspiration index is assessed.

**Author Contributions:** A.T.; methodology, data mining, analysis and modelling, draft reporting, J.C.; analysis and modelling, draft reporting, S.S.; analysis and modelling, draft reporting, D.K.; review analysis and modelling, draft reviewing. All authors have read and agreed to the published version of the manuscript.

**Funding:** The research receives no external funding.

**Institutional Review Board Statement:** Not applicable.

**Informed Consent Statement:** Not applicable.

**Data Availability Statement:** Not applicable.

**Acknowledgments:** We thank both the anonymous reviewers for their suggestions and comments, and we thank the editors for handling the manuscript.

**Conflicts of Interest:** The authors declare no conflict of interest.

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
