# Peer review of "On the Sensitivity of Standardized-Precipitation-Evapotranspiration and Aridity Indexes Using Alternative Potential Evapotranspiration Models"

_hydrology, doi:10.3390/hydrology10030064_

Round 1

Reviewer 1 Report

Review Report

The topic of the paper is interesting and within the scope of the journal. But it needs some modifications for better improvement of the paper. Therefore, I have proposed a “revision” with the following comments:

1.             Write the full name of all the used abbreviations as they come in the paper.

2.             Improve the introduction by highlighting the major difficulties and challenges, and your original achievements to overcome all of these.

3.             What authors were motivated to conduct this type of study? Why they have chosen the six-gauge stations in the California region as a case study?

4.             Add the coordinate frame to Figure 1. Also, improve quality.

5.             What are the advantages of the SPEI over the other drought indexes?

6.             In addition, what is the rationale behind testing the sensitivity of Penman-Monteith, parametric, and Thornthwaite models in SPEI computation?

7.             Check the unit of temperature throughout the paper, and correct it.

8.             Maintain uniform numbers after the decimal place on the y-axis in Figures 2-4. Also, improve the quality of these figures.

9.             Furnish more debate about the practical utility of work in the discussion section.

10.         In conclusion, includes the direction for future works.

Author Response

Dear Reviewer,

We appreciate your efforts for providing your review suggestions .

Please find attached a detailed responses to your concerns.

Reviewer 2 Report

Dear Editors/authors, the article makes an important contribution to technical and scientific knowledge for calculating the aridity index and SPEI. The article is presented in a very succinct and objective way. However, I suggest minor corrections in order to contribute to the improvement of the article.

Small questions:

Are the series presented uninterrupted? Was it not necessary to fill in any gaps?

Among the available stations, why were these six chosen?

Minor fixes:

a) Although well documented and known in the literature, I recommend that the authors be able to explain the values ​​of the coefficients and fixed parameters that were used in the calculations, for example: the value used for the psychometric coefficient, among others in the methodology.

b) Pag 4, line 138: coefficients a, b and c were obtained by the authors for the 6 stations in Table 1, or those estimated by [26] were used.

c) Pag 9, line 234: “It is clear that the parametric model is superior” Can the authors generalize the statement when the present study contemplates only 6 stations? The advantages of the parametric model are clear, but what are the limitations? In what cases would they not be recommended?

d) The description of the results can be improved. Although the authors have listed the main points, it is important to improve the detail of the main results.

e) The legend of the figures needs improvement, making them more self-explanatory. For example, figure 3 is composite, figure 2 as well.

f) The limitations of the approach need to be made explicit, as well as a perspective or suggestions for future approaches.

Author Response

Dear Reviewer,

We appreciate your efforts for providing your review suggestions .

Please find attached a detailed responses to your comments

Round 2

Reviewer 1 Report

No further comments. 

Reviewer 2 Report

Dear editors,

the authors made great efforts to improve the document. In this context, I recommend that the article be accepted for publication.

best regards